# A Bayesian Take on Gaussian Process Networks

**Enrico Giudice**
Dep. of Mathematics and Computer Science
University of Basel, Basel, Switzerland
`enrico.giudice@unibas.ch`

**Jack Kuipers**
Dep. of Biosystems Science and Engineering
ETH Zurich, Basel, Switzerland
`jack.kuipers@bsse.ethz.ch`

**Giusi Moffa**
Dep. of Mathematics and Computer Science, University of Basel, Basel, Switzerland
and Division of Psychiatry, University College London, London, UK
`giusi.moffa@unibas.ch`

## Abstract

Gaussian Process Networks (GPNs) are a class of directed graphical models which employ Gaussian processes as priors for the conditional expectation of each variable given its parents in the network. The model allows the description of continuous joint distributions in a compact but flexible manner with minimal parametric assumptions on the dependencies between variables. Bayesian structure learning of GPNs requires computing the posterior over graphs of the network and is computationally infeasible even in low dimensions. This work implements Monte Carlo and Markov Chain Monte Carlo methods to sample from the posterior distribution of network structures. As such, the approach follows the Bayesian paradigm, comparing models via their marginal likelihood and computing the posterior probability of the GPN features. Simulation studies show that our method outperforms state-of-the-art algorithms in recovering the graphical structure of the network and provides an accurate approximation of its posterior distribution.

## 1  Introduction

Bayesian networks (BNs) are a powerful tool for compactly representing joint distributions and the underlying relationships among a large set of variables [33]. These relationships are described via a directed acyclic graph (DAG), with each node in the graph representing a random variable. The joint distribution of a set of random variables $\mathbf{X} = \{X_1, \dots, X_n\}$ factorizes into conditional distributions for each variable given its parents in the DAG:

$$p(\mathbf{X}) = \prod_{i=1}^{n} p(X_i \,|\, \mathrm{Pa}_{X_i}). \tag{1}$$

The DAG provides a visual description of the dependency structure among the variables, where missing edges encode conditional independence relations among pairs of variables. Thanks to their inherent flexibility and their ability to combine expert knowledge with data, Bayesian networks have been applied to a large range of domains [2].

In the absence of full prior knowledge of the underlying graph, inference on the structure is necessary. For this purpose, a plethora of *structure learning* algorithms have been proposed, which involve either learning the structure from the conditional independence relations in the data (i.e. constraint-based) or assigning a score to each DAG and searching the graph space for high-scoring networks. A recent overview of current algorithms appears in Kitson et al. [23] while a reproducible workflow for

37th Conference on Neural Information Processing Systems (NeurIPS 2023).

benchmark studies is available in Rios et al. [37]. Hybrid methods combining both constraint- and score-based aspects generally offer current state-of-the-art performance.

Typical scoring functions rely on measures of goodness of fit of the graphical model for the given data, such as the Bayesian Information Criterion (BIC) and the Mutual Information Test (MIT) [24, 3]. Another common choice is to use the graph marginal likelihood that we can derive by integrating over the parameter prior on a graphical model. The vast majority of score-based methods implement searches aiming to maximise the scoring function and return one single optimal structure. By complementing the marginal likelihood with a graph prior, one can score graphs according to their posterior distribution and sample from it. By fully characterizing the posterior distribution over graphs it is possible to go beyond optimal point estimates and enable Bayesian model averaging. In this context, approaches have emerged to capture both network and parameter uncertainties across a variety of research areas [27, 1, 32].

Likewise, we focus on Bayesian methods which sample DAGs according to their posterior distribution. Unlike approaches based on maximizing the score function, which return a single graph estimate, Bayesian approaches provide a full posterior distribution over DAGs, which we can use to perform inference on the network's features of interest. However, Bayesian structure inference typically requires a parametric model for the conditional distributions of each variable given its parents in the network to compute the posterior. Variations of the BDe score [21] are the usual choice for discrete-variable networks. Other information theory-based score functions [41, 3] are not Bayesian since they do not estimate a marginal likelihood but rather provide a measure of fitness of a DAG to the data based on the minimum description length principle.

In the case of continuous-variable networks, most of the current research has focused on the linear-Gaussian case, due to the availability of the closed-form BGe score [12, 26]. When relaxing the linear-Gaussian assumption, current approaches fall outside of the Bayesian framework since they do not target a posterior probability for DAGs but rather search the DAG space for a high-scoring network by minimizing a generic loss function. For example, Elidan [6] employs rank correlation as a goodness-of-fit measure between a graph and the observed data; Sharma and van Beek [40] propose modeling the relations among variables via regression splines and scoring the networks via a cross-validated score. Using the formulation by Zheng et al. [52] of the structure learning problem as a constrained continuous optimization, Yu et al. [51] model the conditional distributions via a graph neural network.

In this work, we develop a sampling scheme to perform fully Bayesian structure inference on generic continuous-variable networks with potentially non-linear relations among variables. We follow the strategy of Friedman and Nachman [9] and model the functional dependencies between each variable and its parents via Gaussian process priors. We extend their approach to the Bayesian framework by making structural inference based on the posterior distribution, which also involves treating the hyperparameters of the model as random. After a short review in Section 2 of Bayesian structure inference, Gaussian processes and how they can be used to parameterize BNs, Section 3 proposes a sampling scheme to perform fully Bayesian inference on nonlinear, continuous networks. In Sections 4 and 5 we evaluate and compare our algorithm to existing approaches on simulated and real data.

## 2   Background

### 2.1   Bayesian Structure Inference

When modeling a set of variables $\mathbf{X}$ with a Bayesian network, a typical objective is to estimate the probability of a generic feature of interest $\Psi$. Examples of such a feature are the presence and direction of certain edges, the topological ordering of nodes or conditional independence relations among variables. Given a complete set $D$ of observations of $\mathbf{X}$, we can obtain the posterior probability of $\Psi$ by integrating the probability or presence of the feature over the posterior distribution of graphs:

$$p(\Psi \mid D) = \sum_{\mathcal{G}} p(\Psi \mid \mathcal{G}) \, p(\mathcal{G} \mid D) \,, \quad p(\mathcal{G} \mid D) \propto p(D \mid \mathcal{G}) \, p(\mathcal{G}) \,, \tag{2}$$

wherein a key component is the likelihood of the data integrated over the prior distribution of the parameters $\theta$ of the conditional distributions:

$$p(D \mid \mathcal{G}) = \int p(D \mid \mathcal{G}, \theta) \, p(\theta \mid \mathcal{G}) \, \mathrm{d}\theta. \tag{3}$$

The marginal likelihood $p(D \mid \mathcal{G})$ for a given DAG $\mathcal{G}$ is available in closed form for specific combinations of priors on parameters and likelihood functions [11, 12].

The number of DAGs grows super-exponentially in the number of nodes [38], hence exact posterior inference is still exponentially complex, making it effectively intractable for large networks [47]. Approximate inference for $p(\Psi \mid D)$ is possible if one can obtain samples $\mathcal{G}_1, \ldots, \mathcal{G}_M$ from the posterior distribution over graphs $p(\mathcal{G} \mid D)$.

Markov Chain Monte Carlo (MCMC) methods have successfully tackled the problem of posterior inference in the large and complex space of DAGs. The rationale consists of defining a Markov chain whose stationary distribution is the posterior distribution of interest $p(\mathcal{G} \mid D)$. Madigan et al. [30] suggested a Metropolis-Hastings sampler: at each step, the algorithm proposes a new DAG $\mathcal{G}'$ and it accepts or rejects it according to a probability $\alpha$. The value of $\alpha$ depends on the relative posterior probability $p(\mathcal{G}' \mid D)$ of the proposal graph compared to that of the current DAG:

$$\alpha = \min \left\{ 1, \frac{p(\mathcal{G}' \mid D)\, Q(\mathcal{G}, \mathcal{G}')}{p(\mathcal{G} \mid D)\, Q(\mathcal{G}', \mathcal{G})} \right\} \tag{4}$$

where $Q(\cdot, \cdot)$ are the transition probabilities from one DAG to another. In its original formulation, the algorithm proposes a new graph by adding or deleting one edge from the current DAG, with $Q$ uniformly distributed in the set of neighbouring DAGs (including the current one).

Several refinements to the original method have been made over the years, providing for example more efficient proposal distributions [14, 20, 44]. The order MCMC approach by Friedman and Koller [8] introduced an important variation by sampling in the space of node orderings, which is smaller and smoother than the space of DAGs. Because of these attractive properties of the sample space, the order MCMC algorithm achieves superior mixing and convergence results compared to regular structure MCMC. Sampling orders however introduces a bias in the resulting posterior since node orders do not induce a uniform coverage in the DAG space [7]. The partition MCMC algorithm [25] corrects this bias in order MCMC by sampling in the larger space of ordered partitions of nodes to achieve unbiased samples from the posterior.

Current state-of-the-art methods rely on conditional independence tests to obtain a first coarse estimate of the structure to use as a starting point for a score-based method [48]. More sophisticated approaches involve iteratively expanding the search space to correct for errors in its initial estimate [28]. To achieve efficient posterior inference [49, 28] in the selected search space it helps to precompute all combinations of scores potentially needed in the chain so it can run at a very low computational cost.

Key to enabling efficient DAG sampling is *decomposability*, a property ensuring that we can express the posterior probability of a DAG $\mathcal{G}$ as a product of local scores that depend only on each variable and its parents in $\mathcal{G}$:

$$p(\mathcal{G} \mid D) \propto \prod_{i}^{n} S(X_i, \mathrm{Pa}_{X_i}^{\mathcal{G}} \mid D). \tag{5}$$

Decomposability guarantees that we need to recompute only those scores whose parent sets have changed, or we can precompute all needed combinations of scores efficiently [28].

## 2.2 Gaussian Processes

Gaussian processes are a flexible tool used in machine learning for regression and classification tasks. Formally, a Gaussian Process (GP) is a distribution over functions such that every finite collection of its function values $\{f(x_1), f(x_2), \ldots, f(x_k)\}$ has a multivariate Gaussian distribution [36]. A GP is therefore fully specified by its mean function $m(x) = \mathbb{E}[f(x)]$ and covariance function $k(x, x') = \mathrm{Cov}[f(x), f(x')]$. Due to their ability to model a large range of functional behaviours, GPs find common use as priors over regression functions $f(X) = \mathbb{E}(Y \mid X)$. A common GP regression model assumes independent Gaussian additive noise:

$$Y = f(X) + \varepsilon, \qquad f(X) \sim \mathrm{GP}(0, k(x, x')), \qquad \varepsilon \sim \mathcal{N}(\mu, \sigma^2). \tag{6}$$

Notably, GP models admit a closed-form expression of the marginal likelihood, in this case, the likelihood of the $N$ observations $y$ marginalised over the prior distribution of $f$:

$$p(y) = (2\pi)^{-\frac{N}{2}} \left| K + \sigma^2 I \right|^{\frac{1}{2}} \exp\left( -\frac{1}{2}\, (y - \mu)^{\top} (K + \sigma^2 I)^{-1} (y - \mu) \right) \tag{7}$$

where $K$ is the $N \times N$ Gram matrix $K_{ij} = k(x_i, x_j)$.

## 2.3 Gaussian Process Networks

Gaussian process networks (GPNs) refer to Bayesian networks whose conditional distributions are modeled via Gaussian process priors [9]. The structural equation model defining the distribution of each variable $X_i$ given its parents in a GPN is

$$X_i = f_i(\text{Pa}_{X_i}) + \varepsilon_i \tag{8}$$

where $\varepsilon_i$ has a normal distribution $\mathcal{N}(\mu, \sigma^2)$ independent of the data and a Gaussian process prior is placed on the function $f_i$. Thanks to the nonparametric nature of GPs, the model in (8) can capture a wide range of functional dependencies between variables while maintaining the closed-form expression (7) for the marginal likelihood.

The covariance function $k$ of the Gaussian process is typically parameterized by a set of hyperparameters $\theta$ which determine the behaviour of samples of $f_i$ such as smoothness, shape or periodicity. When these hyperparameters (together with the noise mean $\mu$ and variance $\sigma^2$) are unknown they are typically learned by maximizing the marginal likelihood. Since the marginal likelihood and its derivatives are available in closed form, the optimization can be performed efficiently via gradient ascent [36]. The GPN model enables structure learning of networks on continuous variables without the need to make strict parametric assumptions on the distributions of the variables. In scenarios even with low sample sizes, Friedman and Nachman [9] have shown that searching for the highest-scoring structure can accurately reconstruct the underlying DAG under different functional relationships.

When estimating the score of a GPN, the common approach to learning the hyperparameters by simple maximization can however lead to problematic estimates, since many local optima may exist [4]. In addition, the resulting plug-in estimate of the marginal likelihood would not correctly reflect the uncertainty in the data of the full posterior. Bayesian model averaging provides a natural solution by integrating over the prior distribution of the hyperparameters to obtain a true marginal likelihood $p(D \mid \mathcal{G})$, which we can then use to perform posterior inference on the structure.

# 3 Bayesian Structure Inference for GPNs

In this section, we describe a method to sample from the posterior distribution $p(\Psi \mid D)$ of a generic feature $\Psi$ of a GPN for continuous, non-linear data. To implement a fully Bayesian approach, we place priors over the hyperparameters $\boldsymbol{\theta}$ of the kernel function and the Gaussian noise.

$$\begin{aligned}
X &= f(\text{Pa}_X) + \varepsilon, \quad \varepsilon \sim \mathcal{N}(\mu, \sigma^2) \\
f &\sim \text{GP}(0, k_{\boldsymbol{\theta}}(.,.)) \\
\boldsymbol{\theta} &\sim \pi(\boldsymbol{\theta}), \quad \mu, \sigma \sim \pi(\mu, \sigma).
\end{aligned} \tag{9}$$

The priors ensure that the uncertainty in the functional relationships between each variable and its parents is fully accounted for. On the other hand, a maximum likelihood approach to estimating the hyperparameters could yield overly confident score functions and in turn misrepresent the posterior $p(\mathcal{G} \mid D)$. Under the GPN model, the score remains decomposable, a necessary condition for its efficient evaluation. As in Section 2.1, making inference about network features $\Psi$ of interest is possible by sampling graphs from the posterior:

$$p(\Psi \mid D) \approx \frac{1}{M} \sum_{j=1}^{M} p(\Psi \mid \mathcal{G}_j), \quad \mathcal{G}_j \sim p(\mathcal{G}_j \mid D). \tag{10}$$

Sampling graphs hinges on computing the scores of all variables given different combinations of their possible parent sets (see equation 5).

Let $\Theta = \{\mu, \sigma, \boldsymbol{\theta}\}$ be the $d$-dimensional set of hyperparameters for a given node $X$ and its parents $\text{Pa}_X$. Unless stated otherwise, throughout the rest of the text we assume a uniform prior over all structures $p(\mathcal{G}) \propto 1$. The score function is then the likelihood (7) of the observations $x$ marginalized with respect to the hyperparameter priors:

$$S(X, \text{Pa}_X) = \int p(x \mid \text{Pa}_X, \Theta) \, \pi(\Theta \mid \text{Pa}_X) \, d\Theta. \tag{11}$$

If a variable $X$ has no parents then the Gram matrix of the kernel is zero and the score function reduces to a Gaussian marginal likelihood. Since the above score function is generally intractable,

**Algorithm 1** GP network sampling scheme

---
    **Input** Data $D$ of $n$ variables, feature of interest $\Psi$
    **Output** Posterior probability of the feature $p(\Psi \,|\, D)$
1: **for** $j \in \{1, ..., M\}$ **do**
2:     Sample DAG $\mathcal{G}_j$ according to its Laplace approximate posterior $q(\mathcal{G}_j \,|\, D)$.     ▷ Eq. (13, 5)
3:     **for** $i \in \{1, ..., n\}$ **do**
4:         Compute $S(X_i, \mathrm{Pa}_{X_i}^{\mathcal{G}_j})$ via MC estimation.     ▷ Eq. (12)
5:     Compute posterior $p(\mathcal{G}_j \,|\, D)$.     ▷ Eq. (5).
6: Compute posterior probability of $\Psi$ via importance sampling.     ▷ Eq. (14)

---

one option is to use Monte Carlo (MC) approaches such as bridge sampling [31] to approximate it. Bridge sampling employs a Gaussian proposal distribution $g$ and a bridge function $h$ chosen to minimize the MSE of the resulting estimator. The bridge sampling estimator of (11) is then defined as

$$S(X, \mathrm{Pa}_X) \approx \frac{\frac{1}{N_1} \sum_{i=1}^{N_1} p(x \,|\, \mathrm{Pa}_X, \Theta_i)\, \pi(\Theta_i \,|\, \mathrm{Pa}_X)\, h(\Theta_i \,|\, \mathrm{Pa}_X)}{\frac{1}{N_2} \sum_{j=1}^{N_2} g(\Theta_j^* \,|\, \mathrm{Pa}_X)\, h(\Theta_j^* \,|\, \mathrm{Pa}_X)}, \tag{12}$$

$$\Theta_i \sim g(\Theta \,|\, \mathrm{Pa}_X), \quad \Theta_j^* \sim p(\Theta \,|\, x, \mathrm{Pa}_X), \quad i = 1, ..., N_1, \quad j = 1, ..., N_2.$$

The estimator is also a function of samples from the posterior of the hyperparameters $p(\Theta \,|\, x, \mathrm{Pa}_X)$, which can easily be obtained via MCMC sampling. One can show that other approaches such as importance sampling or harmonic mean are special cases of bridge sampling [18]. MC methods based on bridge sampling provide consistent estimators for the marginal likelihood but may be biased for finite samples [50]. Nonetheless, such methods can become computationally expensive in high dimensions (i.e. for large parent sets).

Since sampling DAGs via MCMC requires computing a large number of scores of potential parent sets, which may not all be represented in the final sample, we avoid computing these expendable scores by first running the MCMC algorithm using a Laplace approximation of the score (11) around the MAP value of $\Theta$:

$$S_{\mathrm{L}}(X, \mathrm{Pa}_X) = p(x \,|\, \mathrm{Pa}_X, \tilde{\Theta})\, \pi(\tilde{\Theta} \,|\, \mathrm{Pa}_X)\, \frac{(2\pi)^{d/2}}{|H|^{1/2}} \tag{13}$$

with $\tilde{\Theta} = \underset{\Theta}{\mathrm{argmax}}\, p(x \,|\, \mathrm{Pa}_X, \Theta)\, \pi(\Theta \,|\, \mathrm{Pa}_X)$, and $H_{ij} = -\frac{\partial^2 p(x \,|\, \mathrm{Pa}_X, \Theta)\, \pi(\Theta \,|\, \mathrm{Pa}_X)}{\partial \Theta_i \partial \Theta_j}\bigg|_{\Theta = \tilde{\Theta}}$.

We denote the resulting posterior probability of a DAG $\mathcal{G}$ from this Laplace approximated score as $q(\mathcal{G} \,|\, D)$ to distinguish it from the true posterior $p(\mathcal{G} \,|\, D)$.

The Laplace approximate score provides an approximation of the posterior at a lower computational cost, speeding up considerably the running time of the MCMC algorithm used to sample graphs. After sampling $M$ graphs in the first step with the Laplace approximate score, we can make inference with respect to the true posterior by re-computing the scores and performing importance sampling. To estimate the posterior probability of a feature of interest $\Psi$ via importance sampling we evaluate

$$p(\Psi \,|\, D) \approx \frac{\sum_{j=1}^{M} p(\Psi \,|\, \mathcal{G}_j)\, w_j}{\sum_{j=1}^{M} w_j}, \quad w_j = \frac{p(\mathcal{G}_j \,|\, D)}{q(\mathcal{G}_j \,|\, D)} \tag{14}$$

where $p(\mathcal{G}_j \,|\, D)$ and $q(\mathcal{G}_j \,|\, D)$ are the posterior probabilities of DAG $\mathcal{G}_j$ computed respectively with the bridge sampling MC estimate (12) and the Laplace approximation (13) for the score. The two posteriors do not need to be normalized for valid posterior inference on $\Psi$. This is because the normalizing constants in the importance sampling weights (14) cancel out. The procedure is summarized as pseudo-code in Algorithm 1.

Re-computing the score functions in a second step and implementing importance sampling is computationally advantageous compared to running the MCMC algorithm directly with the MC estimates of the scores. The advantage is simply due to the number of unique scores in the final chain being much lower than those evaluated or needed during the chain itself (see figure A.5 in the supplementary material). Regardless of the MCMC algorithm used, we expect a substantial improvement in run-time compared to using the MC scores directly.

## 3.1 Implementation Details

Bayesian inference and optimization of the hyperparameters were performed via the Stan interface RStan [43]. The library offers a highly efficient C++ implementation of the No U-turn sampler [22], providing state-of-the-art posterior inference on the hyperparameters. We performed MC estimation of the marginal likelihood via the bridgesampling package [19], which can easily be combined with fitted Stan models.

In the implementation of Algorithm 1 we use order or partition MCMC to generate the network samples $q(\mathcal{G}_j \mid D)$; the procedure can however accommodate a variety of sampling methods, as long as they result in samples from the posterior. Given its good performance in benchmarking studies [37], we use the recent BiDAG [45] hybrid implementation for MCMC inference on the graph. The hybrid sampler requires an initial search space which is then improved upon to correct for possible estimation errors [28]. As initial search space we take the output of the dual PC algorithm [13]. For the bridge sampling estimator (12) of the marginal likelihood we used $N_1 = N_2 = 300$ particles from the proposal and posterior distribution over the hyperparameters. The proposal function $g$ was set to a normal distribution, with its first two moments chosen to match those of the posterior distribution. R code to implement Algorithm 1 and reproduce the results in Sections 4 and 5 is available at https://github.com/enricogiudice/LearningGPNs.

## 3.2 Score Equivalence

Without any assumptions on the functional form of structural models, observational data cannot generally distinguish between any two DAGs within the same Markov equivalence class [34], since the joint distribution always factorizes according to either DAG. Scoring functions like the BGe that assign the same score to all DAGs in the same class satisfy score equivalence. Imposing parametric models on the local probability distributions of a Bayesian network may however break score equivalence. Specifically for GPNs, an alternative factorization may not admit a representation that follows the structural equation model in (8). Consequently, DAGs belonging to the same Markov equivalence class may display different scores and become identifiable beyond their equivalence class. For this reason, the GP score generally differentiates models according to the direction of any of their edges.

Importantly, the identifiability of otherwise score equivalent DAGs is a direct consequence of the assumptions underpinning the GPN model. Because the functions $f_i$ are not generally invertible, the likelihood in (7) will assign higher values to functional dependencies that admit the GPN structural equation model. Further, Peters et al. [35] demonstrated that the asymmetry distinguishing between Markov equivalent factorizations holds beyond the case of non-invertible functions. Indeed, with the exception of the linear case, all structural models that take the form in (8) with additive Gaussian noise violate score equivalence and may identify a DAG beyond its Markov equivalence class. GPNs may encompass cases where equivalent DAGs are indistinguishable, such as when the functions $f_i$ are linear. In this case, the computed scores of equivalent DAGs will be similar, as long as the GP prior allows learning sufficiently linear relations. The numerical experiments in supplementary Section A.1 show that the GP-based score function displays near score equivalent behaviour when the data come from a joint Gaussian distribution, coherently with theoretical considerations [35].

## 4 Experimental Results

We evaluate the Bayesian GP network inference scheme on data generated from known random networks with $n = 10$ nodes. The DAG structures are constructed from an Erdős-Rényi model, where each node has an independent probability of $0.2$ to be connected with another with a higher topological ordering. For every node $X_i$ in each randomly generated network, we then sample 100 observations as a non-linear function of its parents. The nonlinear data are generated by transforming the parents' instances using a weighted combination of six Fourier components

$$X_i = \sum_{j \mid X_j \in \mathrm{Pa}_{X_i}} \beta_{i,j} \left\{ w_{i,j,0} \, X_j + \sum_{k=1}^{6} \Big[ v_{i,j,k} \sin\left(k X_j\right) + w_{i,j,k} \cos\left(k X_j\right) \Big] \right\} + \epsilon_i. \quad (15)$$

The weights $v_{i,j,k}$ and $w_{i,j,k}$ are sampled from a Dirichlet distribution with concentration parameters equal to seven positive exponentially decreasing values $\gamma_k = \frac{e^{-k/\lambda}}{\sum_k e^{-k/\lambda}}$, for $k = \{0, ..., 6\}$. The

parameter $\lambda$ controls the rate of exponential decay, with values of $\lambda$ close to zero providing mostly linear effects between variables and higher values resulting in increasingly non-linear relationships.

The edge coefficients $\beta_{i,j}$ determine the strength of the dependencies between $X_i$ and its parents and are sampled from a uniform distribution on $(-2, -\frac{1}{2}) \cup (\frac{1}{2}, 2)$; the noise variable $\epsilon_i$ has a standard normal distribution. Instances of root nodes (nodes without parents) are also sampled from a standard normal. The linear-Gaussian case corresponds asymptotically to $\lambda = 0$; in this case, we set the weights $v_{i,j,k}$ and $w_{i,j,k}$ to zero except for $w_{i,j,0}$.

We compare all structure learning algorithms in terms of structural Hamming distance (SHD) [48], which compares estimated graphs $\mathcal{G}$ with the ground truth graph $\mathcal{G}^*$. Following Lorch et al. [29], we compute the average SHD of the samples weighted by the posterior:

$$\mathbb{E}-\mathrm{SHD}(p, \mathcal{G}^*) \coloneqq \sum_{\mathcal{G}} \mathrm{SHD}(\mathcal{G}, \mathcal{G}^*) \, p(\mathcal{G}|D). \tag{16}$$

The $\mathbb{E}-\mathrm{SHD}$ summarizes how close the estimated posterior is to the true DAG; lower values are therefore desirable. It is however not a direct measure of the similarity of the estimated posterior to the true posterior.

## 4.1 Choice of Priors

Different choices of kernels for the GP prior result in different behaviours of the conditional expectation $f$ in equation (9) of a variable $X$ given its parents $\mathrm{Pa}_X$. A simple model employs an additive kernel

$$k(.,.) = \sum_{i=1}^{|\mathrm{Pa}_X|} k_{\theta_i}(.,.) \tag{17}$$

which corresponds to modeling each variable $X$ as a sum of the individual contributions of each of its parents. The additive model serves to reduce the computational burden of calculating the scores by keeping the number of parameters as small as possible while preserving non-linearity in the model. The approach can however easily accommodate more complex relationships at a higher computational cost, such as an additive kernel with all first-order interactions [5]:

$$k(.,.) = \tau_1 \sum_{i=1}^{|\mathrm{Pa}_X|} k_{\theta_i}(.,.) + \tau_2 \sum_{i=1}^{|\mathrm{Pa}_X|} \sum_{j=i+1}^{|\mathrm{Pa}_X|} k_{\theta_i}(.,.) \, k_{\theta_j}(.,.) . \tag{18}$$

For each parent $Z \equiv (\mathrm{Pa}_X)_i$ we used a squared exponential kernel function $k_{\theta_i}(z, z') = \exp\left(-\frac{||z-z'||^2}{2\theta_i^2}\right)$, with each $\theta_i$ measuring the degree of non-linearity along the $Z$-th dimension. The kernel function has unit variance since the data are always normalized in the structure learning process. We assign the following independent prior distributions to the hyperparameter set $\Theta = \{\mu, \sigma, \theta_i : i \in 1, \dots, |\mathrm{Pa}_X|\}$ of the GPN model (9):

$$\theta_i \sim \mathrm{IG}(2,2), \quad \mu \sim \mathcal{N}(0,1), \quad \sigma \sim \mathrm{IG}(1,1). \tag{19}$$

The inverse-gamma priors for each lengthscale $\theta_i$ and the noise variance suppress values near zero, which in either case would result in overfitting and degenerate behaviour of samples of $f$ [15]. The additional parameters $\tau_1$ and $\tau_2$ of the kernel function (18) were assigned independent $\mathrm{IG}(1,1)$ priors.

## 4.2 Results

We compare two different versions of the GP-based structure learning scheme: the first one employs the order MCMC algorithm for sampling DAGs, and the second uses partition MCMC. We then compare both versions of the GPN sampling scheme with order and partition MCMC with the BGe score [28]. To reduce the computational burden of the simulations, we employ the simpler additive kernel (17) for the GP score.

As an additional benchmark, we include the DiBS+ algorithm [29], which models the adjacency matrix probabilistically, using particle variational inference to approximate a posterior over structures. In the simulations, we parameterize DiBS+ by a neural network with one hidden layer with 5 nodes. We also consider methods which rely on the constraint-based PC algorithm [42], which

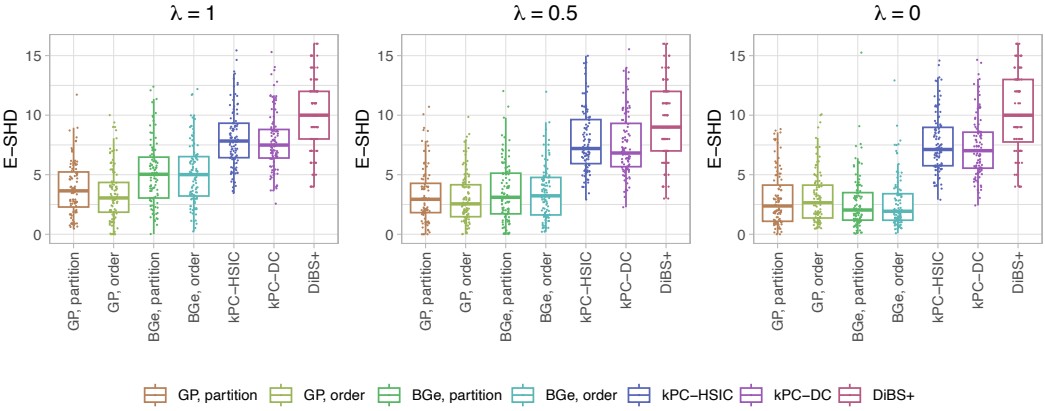

Figure 1: Distribution of $\mathbb{E}-$SHD values for all the different algorithms. $\lambda = 0$ corresponds to linear-Gaussian data while higher values increase the degree of non-linearity of the relations among variables.

learns a network by testing for conditional independence among the variables. Since these are not Bayesian methods, we bootstrap the data to obtain different graph estimates via the PC algorithm [10]. To account for the non-linear dependencies between the variables we apply the kernel PC (kPC) algorithm [17] to the resampled data using two different independence tests: HSIC [16] and distance correlation [46].

Figure 1 shows, for three values of $\lambda$, the distribution of $\mathbb{E}-$SHD values for the different algorithms over the 100 generated structures. For non-linear data ($\lambda = 1$), both versions of the GP samplers outperform existing methods: the median $\mathbb{E}-$SHD for either of our algorithms is equal or lower than the bottom quartile of the BGe samplers. For linear ($\lambda = 0$) and slightly non-linear data ($\lambda = 0.5$), the GPN sampler performs competitively with the state-of-the-art BGe score-based sampling algorithms. All $\mathbb{E}-$SHD values are computed with respect to DAGs; the results comparing completed partially directed acyclic graphs (CPDAGs) are available in figure A.3 in the supplementary material.

Figure 2 displays the run-times for the same methods as in figure 1. The run-times are stable across the different levels of non-linearity parameterized by $\lambda$, with the k-PC algorithms being the most computationally expensive, followed by the GP score-based approach, DiBS+ and finally the BGe score-based MCMC methods which are the fastest.

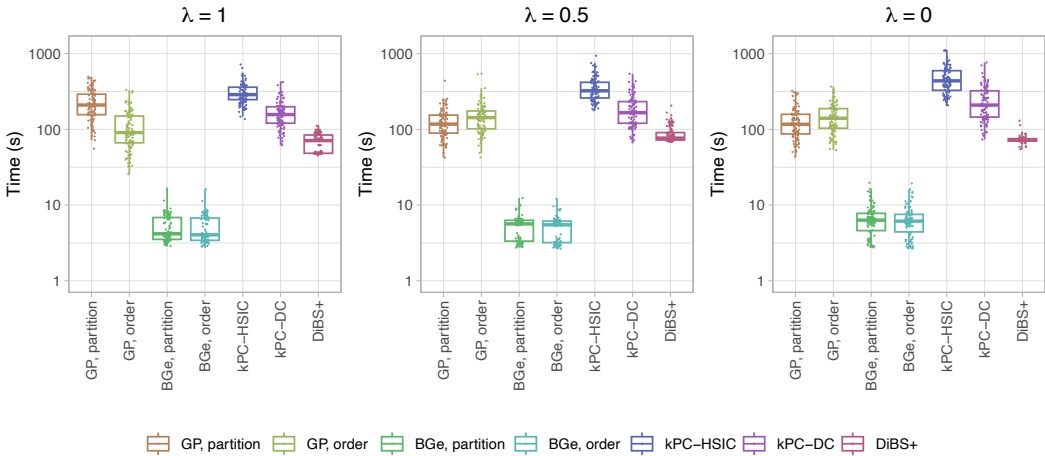

Figure 2: Distribution of run-times for all the different algorithms. $\lambda = 0$ corresponds to linear-Gaussian data while higher values increase the degree of non-linearity of the relations among variables.

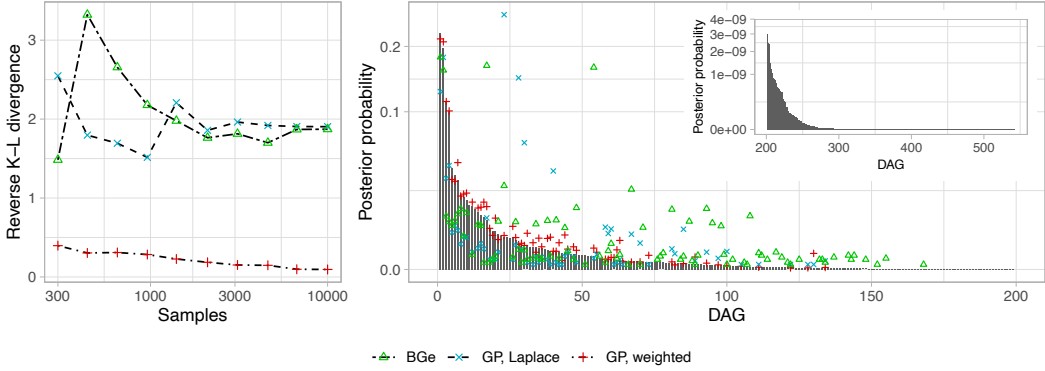

Figure 3: Left: reverse K-L divergence between the true posterior and the BGe posterior (green), the Laplace approximate posterior (blue) and the posterior obtained via importance sampling (red) as a function of the number of sampled DAGs. Right: the true posterior (gray) together with the BGe posterior (green), the Laplace approximate posterior (blue) and the posterior obtained via importance sampling (red). The majority of DAGs have a very low true posterior probability and are therefore never sampled by the MCMC algorithms (see inset).

Besides estimating accurately the structure, our approach can quantify the uncertainty in its estimates via the sampled graphs from the posterior distribution over GPNs. To evaluate the accuracy of the sampling approach in estimating the general posterior over structures, we compare its estimated posterior distribution over DAGs with the "true" posterior, obtained by enumerating every possible structure and computing its score directly with equation (12). Due to the exceedingly large number of DAGs, this approach is only feasible with small structures. The left panel of figure 3 shows the reverse Kullback-Leibler (K-L) divergence between the estimated and true posteriors for a given network with $n = 4$ nodes, as a function of the number of samples $M$ in the partition MCMC algorithm. The estimated ("GP, weighted") posterior probability $p(\mathcal{G} \mid D)$ for a generic DAG $\mathcal{G}$ is obtained by setting $p(\Psi \mid \mathcal{G}_j) = \mathbb{1}_{(\mathcal{G}_j = \mathcal{G})}$ in equation (14). Reverse K-L divergence was chosen as a metric since the algorithms assign a probability of zero to DAGs that were not sampled.

The plot includes the divergence between the Laplace approximate posterior $q(\mathcal{G} \mid D)$ in equation (14) and the true posterior, as well as between the posterior obtained with the BGe score and the true posterior. The divergence of the Laplace approximation is reduced by roughly one order of magnitude by weighting the samples via importance sampling. For reference, the DiBS+ algorithm yields a reverse K-L divergence of 27.8 with 1000 samples, i.e. two orders of magnitude higher than our approach, despite also being allocated longer run-times (see figure A.4 in the supplementary material). Sampling a higher number of graphs with DiBS+ quickly becomes infeasible since its run-time scales quadratically with the number of samples [29], while MCMC sampling scales linearly with $M$. The right panel of figure 3 shows the posterior distributions over the 543 possible DAGs, for $M = 10^4$, obtained after complete enumeration and by sampling with different methods. The plots confirm that as the number of sampled DAGs increases, our approach can accurately reflect the full posterior uncertainty. The results also underline the importance of sampling from the hyperparameters' priors to obtain an accurate representation of the posterior, as the Laplace approximation of the marginal likelihood results in a highly biased approximation even when sampling a large number of DAGs.

## 5 Application on Protein Signaling Networks

We also applied the GP score-based structure inference to the flow cytometry dataset of Sachs et al. [39] to learn protein signaling pathways. The authors provide an accepted consensus network, which we used as reference. We considered the first experimental condition, consisting of 853 single-cell observations of $n = 11$ phosphoproteins and phospholipids in human T cells. The first three columns of table 1 show the performances of all the different algorithms in reconstructing the consensus network. The results include the GP model using the additive kernel (18) with all first-order interactions, denoted as GP$^2$. We measure the different algorithms in terms of the $\mathbb{E}-$SHD,

Table 1: Performance of the different algorithms in reconstructing the consensus network from flow cytometry data. The last two columns show the posterior probabilities of the two features experimentally validated by Sachs et al. [39], where the edge on the left should be present and the one on the right absent; up/down arrows indicate higher/lower values are better.

| | $\mathbb{E}-$SHD $\downarrow$ | $\mathbb{E}-$TP $\uparrow$ | $\mathbb{E}-$FP $\downarrow$ | Erk $\rightarrow$ Akt $\uparrow$ | Erk $\nrightarrow$ PKA $\uparrow$ |
|---|---|---|---|---|---|
| GP, partition | 14.5 | 7.2 | 4.7 | 1 | 0.75 |
| GP, order | 14.6 | 6.8 | 4.4 | 1 | 0.42 |
| $GP^2$, partition | 16.7 | 6.6 | 6.3 | 1 | 0 |
| $GP^2$, order | 16.7 | 6.7 | 6.4 | 1 | 0.34 |
| BGe, partition | 15.9 | 4.3 | 3.2 | 0 | 1 |
| BGe, order | 15.3 | 4.4 | 2.7 | 0.17 | 0.98 |
| kPC-HSIC | 17 | 5.5 | 5.5 | 0.69 | 0.26 |
| kPC-DC | 16.9 | 6 | 5.8 | 0.72 | 0.28 |
| DiBS+ | 16.1 | 5 | 4.2 | 0.45 | 0.7 |

as well as the $\mathbb{E}-$TP and $\mathbb{E}-$FP, the absolute number of TP and FP edges in the DAGs weighted by the posterior, obtained by replacing the SHD in equation (16) by TP or FP, respectively.

One of the benefits of Bayesian structure inference is the possibility of deriving posterior probabilities of specific edges of interest in the network. In their work, Sachs et al. [39] experimentally tested two relationships between the proteins by intervening directly on the cells. By means of small interfering RNA inhibition, they concluded that the inhibition of Ekt has a direct effect on Akt, while there was a weaker and non-significant effect on PKA. We therefore expect an edge Erk $\rightarrow$ Akt but no directed edge (or path) from Erk to PKA in the learned networks. The last two columns of table 1 display the posterior probabilities of these two features according to the different algorithms. GP- and BGe-based methods perform best, with the GP learner correctly assigning the highest probability to the edge Erk $\rightarrow$ Akt, while the lack of the edge Erk to PKA is predicted with a lower probability. The sparser BGe score-based methods assign a lower probability to the first edge, but correctly predict the absence of the second edge.

## 6    Conclusions

In this work, we have proposed a procedure that efficiently performs Bayesian structural inference on Gaussian process networks and estimates the posterior of generic features of interest. Building on the original GPN idea by Friedman and Nachman [9], we now embed it in a fully Bayesian framework. In particular, our approach involves placing priors on the hyperparameters of the score function and sampling DAGs via MCMC. Although more computationally expensive than a greedy search over the DAG space for a high-scoring network, the Bayesian approach allows one to accurately quantify the uncertainty in features of the network. This is made feasible by minimizing the number of scores to compute in an MCMC chain via importance sampling, as well as harnessing the advances in MC and MCMC methods [19, 28, 43, 45] that have been made since the introduction of GP networks.

The flexible nature of GPs allows the corresponding BNs to model a large range of functional dependencies between continuous variables. In the case of linear dependencies, our method remains competitive with state-of-the-art methods and exhibits desirable properties such as (approximate) score equivalence. The versatility of the model is particularly useful in domains where the nature of relations among variables in unknown and strict parametric assumptions on the distributions of the variables want to be avoided. Based on the promising simulation results and the convenient properties of the proposed method, we believe that it holds potential for making accurate inference on the underlying structure of BNs in complex domains.

### Acknowledgements

The authors are grateful to acknowledge partial funding support for this work from the two Cantons of Basel through project grant PMB-02-18 granted by the ETH Zurich.

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
