# A Supplementary Material

## A.1 Measuring Score Equivalence

In this section, we perform a simulation study to measure the degree to which the GP score is capable of inferring the correct direction of the edges in the graph. Following the considerations of Section 3.2, the experiment aims to gauge to what degree the model can correctly identify edge directions for non-linear data while retaining score equivalence in the linear Gaussian case.

We generate $100$ observations following the approach outlined in Section 4 from a "forward" network consisting of a chain of $5$ nodes $X_1 \longrightarrow ... \longrightarrow X_5$. We then compare the score of the forward network with that of a "backward" network, which has all edges reversed. In figure A.1 we then plot the difference in log-score between the forward and backward models as a function of the non-linearity parameter $\lambda$. For each value of $\lambda$, the log-difference in scores was averaged over $100$ runs.

The results show that for roughly linear-Gaussian relationships ($\lambda \approx 0$), score equivalence of the Bayesian GP score holds in expectation, while for larger deviations from the linear-Gaussian model, the score increasingly favors the correct forward DAG. For $\lambda = 0.75$, the score of the forward model is greater than that of the backward model in more than $90\%$ of the simulations.

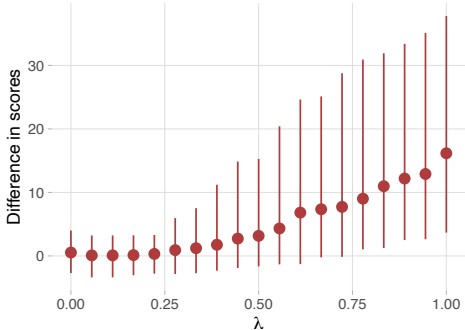

Figure A.1: The median difference in GP log score between the forward and backward model, with $0.1$- and $0.9$-quantiles.

## A.2 Additional Simulation Results

Further to the results in Section 4.2 of the main text, here we present additional metrics comparing the various algorithms' performances. We construct ROC-like curves counting the average number of true positive (TP) and false positive (FP) edges of the highest-scoring graph. We define the true positive edge rate and a modified false positive edge rate as

$$\text{TPR} = \frac{\text{TP}}{\text{P}} \qquad \text{FPRp} = \frac{\text{FP}}{\text{P}} \, . \tag{20}$$

We scale the FPs by the number of positives to obtain measures on a similar scale, since the number of true negative edges can be exceedingly large for DAGs. Different points in the ROC space are produced by varying the hyperparameters of the different structure learning algorithms; the process also allows tuning the hyperparameters of each algorithm to minimize their $\mathbb{E}-\text{SHD}$.

Figure A.2 displays the TPR and FPRp for different values of the hyperparameters for a selection of the considered algorithms. For the kPC algorithm-based methods we vary the parameter controlling the type I error; for the BGe score-based methods we vary the prior on the precision matrix of the Gaussian likelihood. Since the GP score-based algorithms do not rely on any hyperparameters that directly control the sparsity of the resulting graphs, we simply penalize the number of edges in the DAGs via a prior on the graph space. The prior depends on the number of edges according to the following distribution: $p(\mathcal{G}) \propto \exp\{-\gamma \#(\text{edges})\}$. The ROC-like curves are then constructed by averaging the TPR and FPRp over the $100$ runs for each value of the hyperparameters, under the

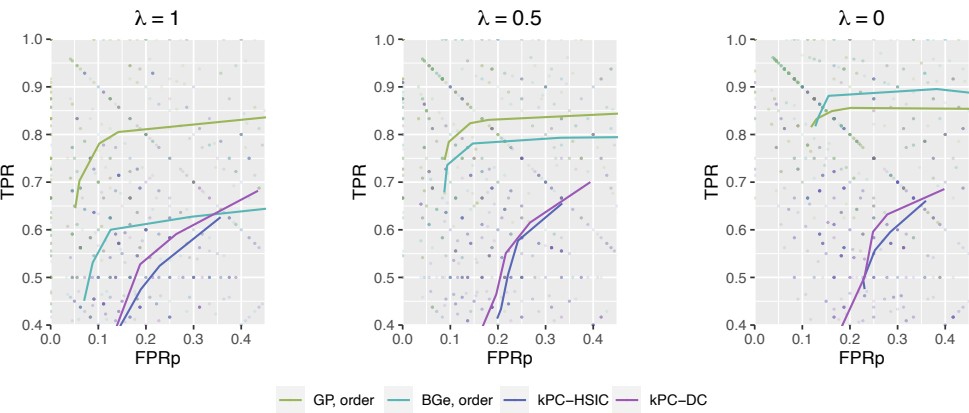

Figure A.2: Average FPRp and TPR values for a selection of the different algorithms. $\lambda = 0$ corresponds to linear-Gaussian data while higher values increase the degree of non-linearity of the relations among variables.

settings described in Section 4. The partition MCMC results are omitted since they are in very close agreement with the output of order MCMC.

According to the ROC-like curves, the two BGe score-based methods perform only slightly better than the GP-based methods for linear data; in all other cases, the GP-based methods achieve both higher TPR values and lower FPRp values compared to all the other benchmarks. The value of the hyperparameters that minimized the $\mathbb{E}-$SHD of each algorithm was used to obtain the plots in figure 1 of the main text.

Figure A.3 shows the distribution of $\mathbb{E}-$SHD values computed with respect to the true and estimated CPDAGs for the same experiments in figure 1; the results are in line with those comparing the DAGs. Cyclic graphs occasionally returned by DiBS+ were discarded.

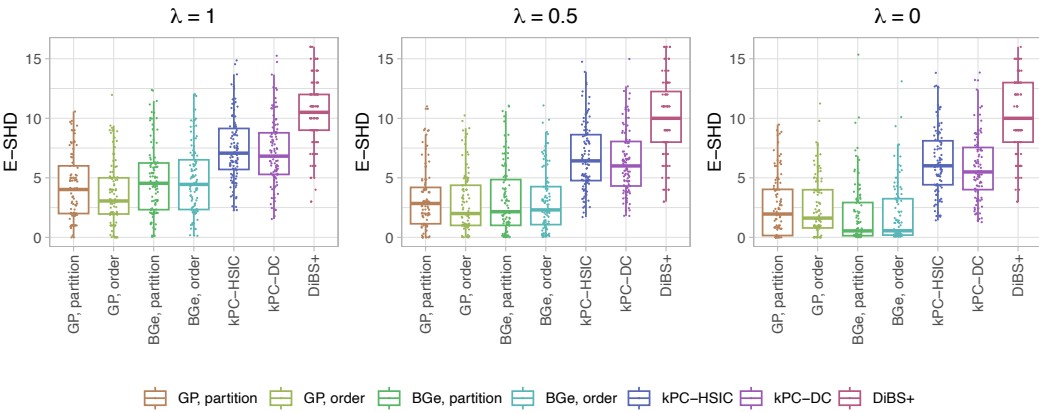

Figure A.3: Distribution of $\mathbb{E}-$SHD values comparing the true and estimated CPDAGs. $\lambda = 0$ corresponds to linear-Gaussian data while higher values increase the degree of non-linearity of the relations among variables.

We performed an additional experiment comparing the ability of the different methods to model the posterior distribution over DAGs as a function of their run-time. Figure A.4 shows the reverse K-L divergence between the "true" posterior (obtained by enumerating every possible structure and computing the score of the GPN via bridge sampling) and the posteriors estimated with the different methods. We used the same 4-node DAG of figure 3, and the different run-times were obtained by increasing the number of samples taken from the posterior for each method. Three methods are compared: partition MCMC with the BGe score, DiBS+ and the approach of using partition MCMC with the GP score together with importance sampling.

The results show that as the number of sampled DAGs increases, the GP-based method can success-fully represent the posterior distribution. The BGe score-based approach and DiBS+ are not however able to reach low K-L values, even as the number of samples is increased and the methods are given additional run-time.

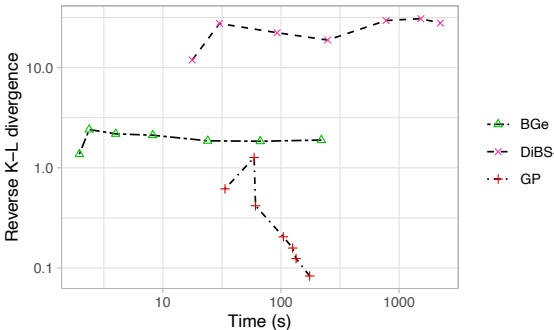

Figure A.4: Reverse K-L divergence between the true posterior and the BGe posterior (green), DiBS+ (pink) and the posterior obtained via the GP score-based approach (red) as a function of run-time.

In figure A.5 we compare the number of score evaluations performed by the different methods when learning networks with $n = 10$ nodes with Algorithm 1. The plot shows that the number of scores in the final MCMC samples (in brown/green) is much lower than the total number of scores evaluated when building the chain (in blue). The importance sampling-based approach therefore leads to a much more efficient procedure than naïvely computing all scores (in blue) with bridge sampling. The number of Laplace approximate scores is the same for order and partition MCMC since all potentially needed scores in the chain are pre-computed following the approach of [28].

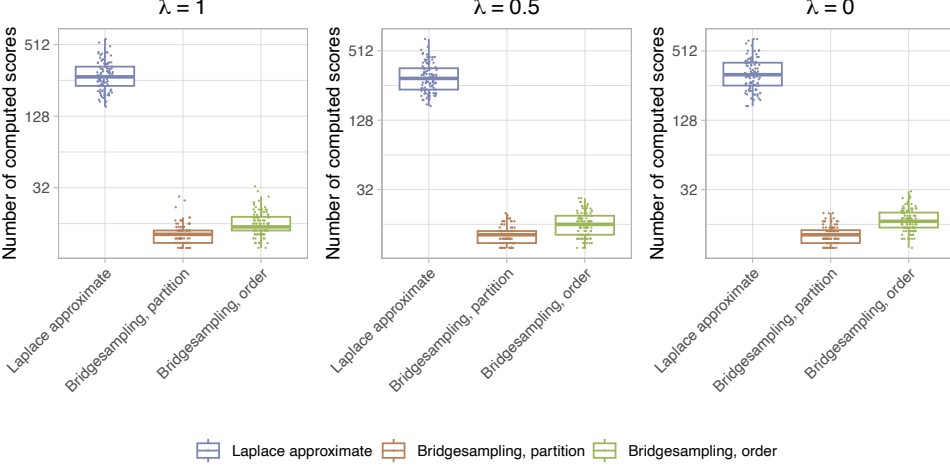

Figure A.5: Distribution of number of scores evaluated by the different methods. In blue is the total number of scores computed via the Laplace approximated posterior when running the MCMC algorithms ("Laplace approximate"), in brown is the number of scores evaluated via bridge sampling for the partition MCMC algorithm ("Bridgesampling, partition") and in green the same quantity for order MCMC. $\lambda = 0$ corresponds to linear-Gaussian data while higher values increase the degree of non-linearity of the relations among variables.

To see to what extent the results are dependent on the choice of prior, we study the effect of different choices of the prior on the lengthscales $\theta_i$. Figure A.6 displays results for three different parameterizations of the inverse-gamma prior for networks with $n = 10$ nodes. The plot shows the performance of the partition and order MCMC version of Algorithm 1 with three different priors: the default $IG(2, 2)$ prior, a $IG(1, 3)$ prior that favors higher values of $\theta$, and a $IG(3, 1)$ prior favoring

lower length scales. The results indicate that varying the prior hyperparameter has a limited effect on the $\mathbb{E}-$SHD values.

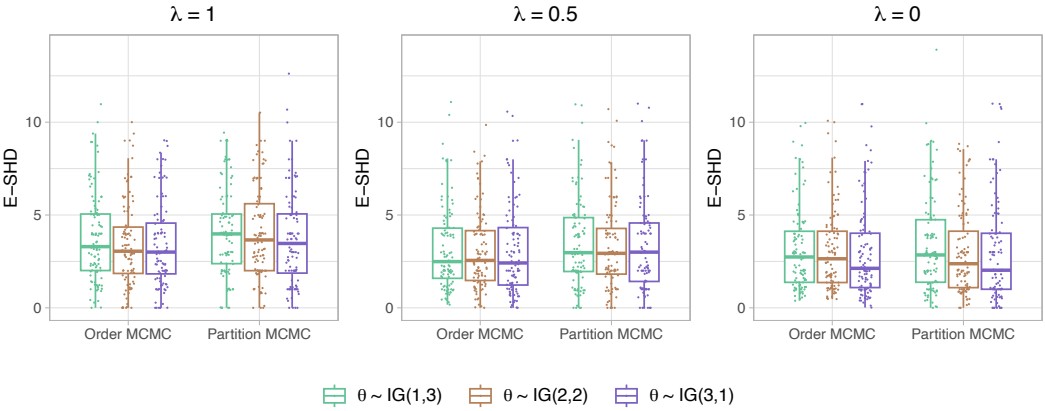

Figure A.6: Distribution of $\mathbb{E}-$SHD values for different choices of the parameterization of the inverse-gamma priors on the lengthscales $\theta_i$. $\lambda = 0$ corresponds to linear-Gaussian data while higher values increase the degree of non-linearity of the relations among variables.

Finally, we repeated the simulations outlined in Section 4 on larger randomly generated DAGs with $n = 15$ nodes. Figures A.7 and A.8 show the distribution of $\mathbb{E}-$SHD values computed respectively with respect to DAGs and CPDAGs. Figure A.9 shows the corresponding run-times needed to run each algorithm. The relative performance of the different methods does not differ compared to the results with lower-dimensional networks.

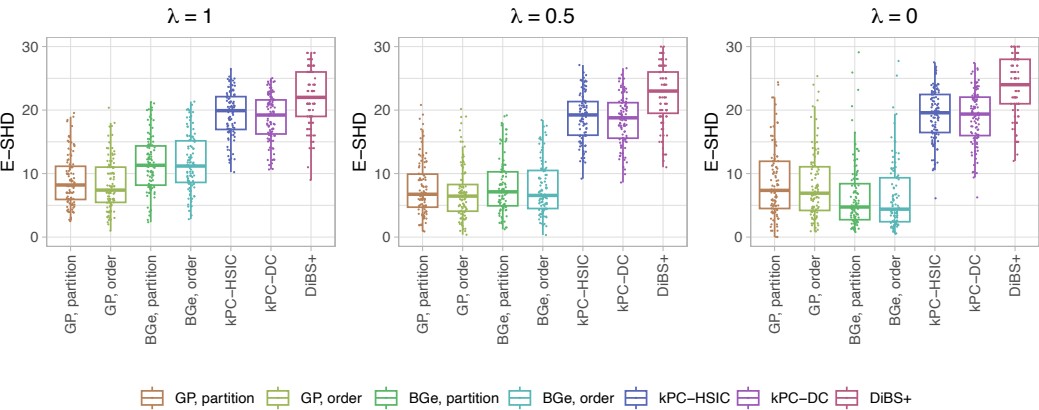

Figure A.7: Distribution of $\mathbb{E}-$SHD for networks with $n = 15$ nodes, computed on the DAG space. $\lambda = 0$ corresponds to linear-Gaussian data while higher values increase the degree of non-linearity of the relations among variables.

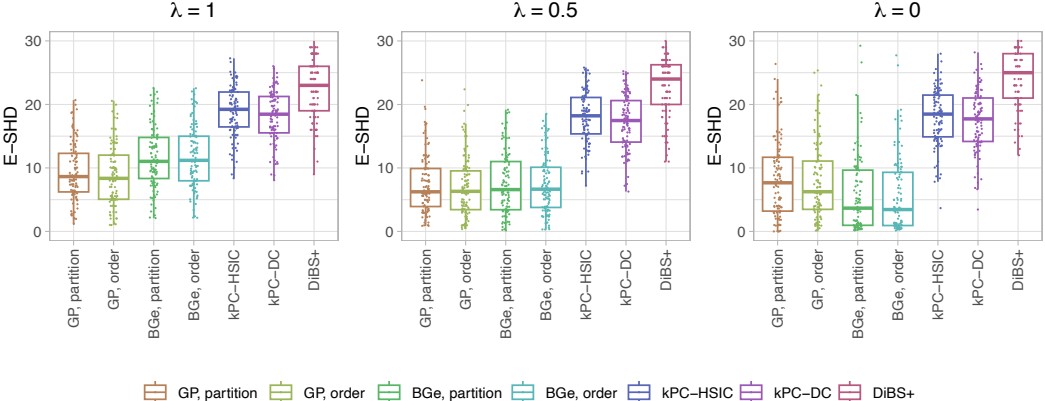

Figure A.8: Distribution of $\mathbb{E}-\text{SHD}$ for networks with $n = 15$ nodes, computed on the CPDAG space. $\lambda = 0$ corresponds to linear-Gaussian data while higher values increase the degree of non-linearity of the relations among variables.

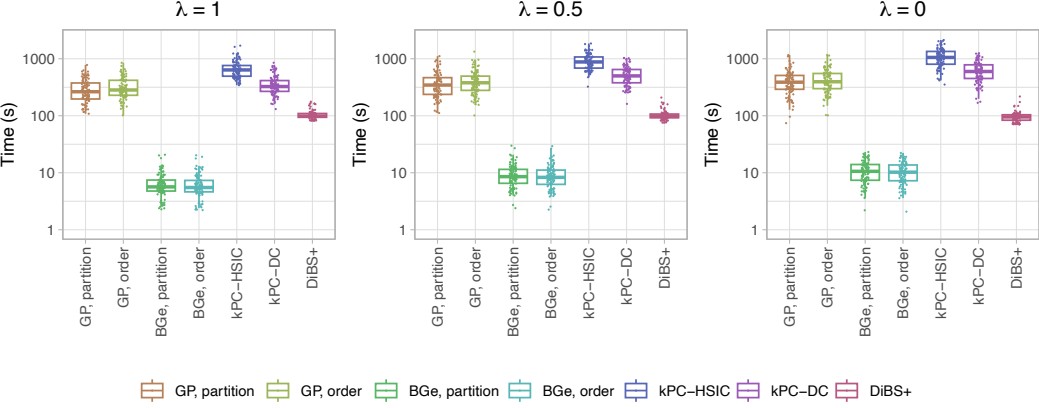

Figure A.9: Distribution of run-times for networks with $n = 15$ nodes. $\lambda = 0$ corresponds to linear-Gaussian data while higher values increase the degree of non-linearity of the relations among variables.