# OpenReview forum: "A Bayesian Take on Gaussian Process Networks"
_NeurIPS.cc/2023/Conference — NeurIPS 2023 poster_

### Official Review · Reviewer_9qjJ · 2023-06-21

**Soundness:** 3 good
**Presentation:** 3 good
**Contribution:** 3 good
**Rating:** 6
**Confidence:** 4

**Summary:**

Gaussian Process (GP) networks are directed graphical models for continuous data, where the function mapping from parent node values to parameters of child node is a Gaussian process. Given the graphical network structure and a dataset of observations, learning the GP model for each node simply reduces to a standard supervised GP regression using parents nodes as input $x$ values and child node as output $y$ values. Likewise marginal likelihood evaluation and hyperparameter optimization/marginalization can use the standard methods for GP regression.

In this work, as I understand, the authors goal is to marginalise over network structures, each network structure yields a different set of parents-child links and each one requires marginalizing GP hyperparameters again, if done naively, this is very very expensive. The authors thus propose methods to speed up this process hence making network structure marginalisation much more computationally efficient. These methods include
- using Metropolis Hasting like algorithms to make small steps through the space of graphs thereby removing the need for excessive recomputation.
- for each graph in the Metropolis Hastings proposal, the authors use a Laplace approximation to marginalize GP hyperparameters instead of full MCMC marginalisation.
- once a dataset of sampled graphs is collected, the expensive MCMC marginalisation is used to weight each sampled graph with importance sampling.



Experiments on both synthetic toy example and a protein interaction network example are presented.

**Strengths:**

- I feel that the proposed methods are very standard approaches and I feel are well justified in this use case and the overall model and algorithm is is well designed.
- personally coming from a traditional GP background, I felt this paper was accessible and easy to digest.

**Weaknesses:**

- Everything from line 101 up until line 159 is focused on model fitting and hyperparameter marginalization for a single GP and, as far as I can tell, has nothing to do with networks, hence the section title felt a bit misleading and rearranging material might make this easier. Equations 8 - 12 could hypothetically be written using $x, y$ instead of $Pa_{X}, X$ and moved to Section 2. Then section 3 could contain a description about how graph structure determines each nodes "local GP training set" and how the model is a summation over GPs over individual training sets. and the fact that there are so many justifies a hyper-parameter approximation.
- Equation (10) appears to be a standard marginal likelihood with an extra layer of marginalisation over kernel hyperparameters (with or without hyperpriors), this is standard practice in GP regression [Code here](https://github.com/rmgarnett/gpml_extensions/) and [this paper](https://proceedings.neurips.cc/paper/2010/hash/4d5b995358e7798bc7e9d9db83c612a5-Abstract.html). The authors cite bridge sampling which I am not personally familiar with, but there exist many, many algorithms to solve this problem (this is not a weakness of the paper as any MCMC algorithm will suffice, but at first glance seems a rather strange choice)

- In Section 4, the authors repeatedly state that $\lambda =0$ removes higher frequencies and is a linear case and so $w_{i,0}$ is the only non-zero coefficient. If I take equation 14 and make the suggested substitution, I have a constant with noise
$$
X_i = \sum_{Z\in Pa_{X_i}} \beta_i w_{i,0} + \epsilon_i
$$
Which implies no dependency between $X$ and $Z$. (The only way I could see a linear relationship between $X$ and $Z$ is if $\sin(jZ)$ for very small $j$ where $\sin()$ is approximately linear near the origin, but given normalised $Z$ and $j=1$ this isn't likely). Hence the $\lambda=0$ case is generating data that ignores graph structure, and the hamming distance is meaningless as the models will learn the empty graph from the data.

**Questions:**

- as mentioned above regarding $\lambda=0$, is my understanding correct? Can the authors comment on this?
- Figure 1: I assume the graph hamming distance shows high "tight" the posterior over graphs is clustered around the true graph, tighter is better, whereas with 100 observations in 10D space, this seems very sparse dataset to learn any complex model and I would expect the posterior to be rather broad. What is the hamming distance of uniformly randomly generated graphs? What is the average hamming distance of the "true" posterior from the true generating graph?

**Limitations:**

- the authors provide an explanation of the Score equivalence issue (which hadn't occurred to me until it was mentioned)

---

> ### Author Rebuttal · Authors · 2023-08-09
>
> We would like to thank the reviewer for their constructive comments and suggestions.
>
> * *as mentioned above regarding $\lambda=0$, is my understanding correct? Can the authors comment on this?*
>
> There is indeed a typo in equation (14), the linear terms should have been included and the index $j$ of the second sum should start at 1 instead:
> $$ X_i \\, = \sum_{Z \\, \in \\, \textrm{Pa}_{X_i}} \beta_0 \\, w\_{i,0} \\,Z \\,+\\, \\left\\{\sum\_{j=1}^6 \beta_i \\,v\_{i,j} \sin{(j Z)} \\,+\\, \beta_i\\, w\_{i,j} \cos{(j Z)} \\right\\} \\,+\\, \epsilon_i. $$
> It was however coded correctly in the experiments. We thank the reviewer for this careful observation, and will update equation (14) in the manuscript.
> ***
> * *Figure 1: I assume the graph hamming distance shows high "tight" the posterior over graphs is clustered around the true graph, tighter is better, whereas with 100 observations in 10D space, this seems very sparse dataset to learn any complex model and I would expect the posterior to be rather broad. What is the hamming distance of uniformly randomly generated graphs? What is the average hamming distance of the "true" posterior from the true generating graph?*
>
> Tighter is indeed better, but only up to a point, because as noted the posterior is spread and its average distance is non-zero. To check convergence to the true posterior we need measures like the K-L divergence in Figure 2 of the main text, but this requires enumerating all DAGs and computing their scores. The current best complexity for exact computations scales like 3^(# of nodes) times the score computation time, and is only feasible for small graphs (as in Figure 2). The number of DAGs with 10 nodes is over 4 quintillion, making it infeasible to compute the exact E-SHD of the true posterior distribution. However, since we have samples from the posterior, the “GP partition” results in Figure 1 of the main text are Monte Carlo estimates of the average Hamming distance between the true posterior and the true generating graph (and the “BGe partition” for a linear-Gaussian model). Exact computations for the uniform distribution over DAGs are cheaper (there is no score component) and MC estimates are simple, so this could be added, but as an average DAG with 10 nodes has more than 20 edges, the distances would, as we might expect, be high compared to the scale of the figure.
> ***
> * *the authors provide an explanation of the Score equivalence issue (which hadn't occurred to me until it was mentioned)*
>
> We’re glad the reviewer appreciated this aspect, as it is often overlooked. It is unclear to us why this is a limitation though.

---

### Official Review · Reviewer_sMvJ · 2023-07-03

**Soundness:** 2 fair
**Presentation:** 3 good
**Contribution:** 2 fair
**Rating:** 3
**Confidence:** 5

**Summary:**

This paper proposes a Bayesian structure learning of the GPNs framework that is claimed to be less computationally. To address this, the approach presented in this work utilizes Monte Carlo and importance sampling to sample from the posterior distribution of network structures. This approach compares models using their marginal likelihood and computes the posterior probability of GPN features. Some simulations have been discussed.

**Strengths:**

- This paper studies a rather important problem.
- It is easy to follow the thoughts.

**Weaknesses:**

- The contribution of the paper is not clearly justified and seems moderate at best. It is a combination of a few approaches (Gaussian processes, Laplace approximation, and Importance sampling).
- The claims of the paper are not completely justified. It provides an approach but does not discuss how the main concerns are addressed. It is not clear how efficient this method is relative to others.
- Some references are missing, for instance:
     * Efficient and scalable structure learning for Bayesian networks: algorithms and applications by Zhu 2020
     * Efficient structure learning of Bayesian networks using constraints by de Campos
     * MCMC algorithms for Bayesian analysis of phylogenetic trees
- Lack of comparison/validation to other state-of-the-art algorithms and also lack of discussion on how efficient this framework is by providing computational intensity analysis.
- Lack of theoretical discussion regarding the convergence and issues rising from Laplace approximation and Importance sampling.
- Lack of validation, sensitivity analysis and comparison to alternative models.

**Questions:**

A few questions that come to mind:
- Laplace approxiamtion is sensitive to the MAP estiamte. as the score s are being approximated this may cause poor approxiamtion and bias and thus it may impact the perfomrance of the subsequenct imporatnce sampling and causing high variance of imporance weights and thus inefficient sampling and inaccurate estmation. could the paper elaborate on this issue.
- Imporatnce sampling can introduce bias if the proposal does not adequetly cover the suppot of the target posterior. Also, high variance of the weights can lead to unstable estiamtes.
- Althogh use of Lapace approxiamtion and imporatance sampling may in general reduce the computaional intensity  but overall performance dependes on model and the size of dataset (specifically when approxiamting the score and so on). It is well-known that as dimension of the netwrok increases or dataset grows larger the computational cost of Laplace approxiamtion and imporatnece samling can be significant. How is the relationship between effective sample size, variabaility of the weights and growth of the netwrok, dataset, and dimentionality justified?

**Limitations:**

- Laplace approximation is only valid if the distribution is Gaussian or close to Gaussian. This assumption can simply be violated.
- Lack of discussion on computational intensity and how this framework overcomes those known computational issues.
- Lack of validation, sensitivity analysis and comparison to alternative models.

---

> ### Author Rebuttal · Authors · 2023-08-09
>
> We would like to thank the reviewer for their comments, and acknowledge the time they took to assess the paper.
> * *The contribution of the paper is not clearly justified and seems moderate at best. It is a combination of a few approaches (Gaussian processes, Laplace approximation, and Importance sampling).*
>
> The objective of our work is addressing an important problem, as recognised also in the review, indeed drawing on technical tools originating in different domains of the statistical inference literature and we combine them effectively in a novel whole. The Bayesian approach accounts for uncertainty both in the parameters and structure, allowing us to compute posterior probabilities of network features in a principled manner.
> ***
> * *The claims of the paper are not completely justified. [...] It is not clear how efficient this method is relative to others.*
>
> The supplementary material already included analyses of the run-times illustrating the computational costs of the procedure, which are expected to be higher compared to simpler methods. The added value is that this is the only method correctly targeting the posterior distribution (as illustrated for example in Figure 2 in the main text).
> ***
> * *Some references are missing, for instance [...]*
>
> The references suggested are not entirely pertinent to the topic covered in the paper. The first two references do not qualify as Bayesian methods, since they do not provide a posterior distribution over DAGs and/or do not compute the marginal likelihood of the data, opting for a generic score function instead. The last reference concerns phylogenetic trees and is not directly related to our problem of Bayesian network structure learning. However, we agree that to better embed the work in the literature, adding some text on non-Bayesian point estimate approaches, and then modifying the current sentence to better emphasize the Bayesian side would work well, and we will modify accordingly.
> ***
> * *Lack of comparison/validation to other state-of-the-art algorithms and also lack of discussion on how efficient this framework is by providing computational intensity analysis.*
>
> There are few methods that currently are able to perform Bayesian structure learning for BNs with non-linear, continuous data. Again, as for the references, we have to distinguish between Bayesian and non-Bayesian approaches. We of course welcome concrete suggestions for other algorithms to include. As for the computational intensity analysis, we already examined this in Figures A.3, A.5 and A.8 in the supplementary material.
> ***
> * *Lack of validation, sensitivity analysis and comparison to alternative models.*
>
> We compared the performance of our method to five different state-of-the-art algorithms, and provided a comparison of run-times in Figures A.3, A.5 and A.8 in the supplementary material. Furthermore, we have now added new figures in the additional pdf in the global response, which include an analysis of our method’s performance for different choices in the prior hyperparameters.
> ***
> * *Laplace approximation is only valid if the distribution is Gaussian or close to Gaussian. This assumption can simply be violated.*
>
> The Laplace approximation is indeed only exact if the posterior distribution is Gaussian; we are however not relying directly on this approximation to compute the marginal likelihoods. We are using the Laplace approximation as a proposal distribution for subsequent importance sampling. The exact posterior is then computed by marginalizing the likelihood with respect to the prior using bridge sampling. The results so obtained finally provide valid samples.
> ***
> * *Laplace approxiamtion is sensitive to the MAP estiamte. as the score s are being approximated this may cause poor approxiamtion and bias and thus it may impact the perfomrance of the subsequenct imporatnce sampling and causing high variance of imporance weights and thus inefficient sampling and inaccurate estmation.*
>
> * *Imporatnce sampling can introduce bias if the proposal does not adequetly cover the suppot of the target posterior. Also, high variance of the weights can lead to unstable estiamtes.*
>
> The same Laplace approximation is used both in the target for graph sampling and in the importance sampling. Even if the approximation is poor, these terms “cancel” out and do not induce bias, though a poor approximation may increase variance. A high variance in the weights will only be observed when there is a large discrepancy between the proposal and true posterior. Using a Laplace approximation as a proposal distribution for importance sampling is commonplace in the literature (see e.g. Kuk, 1998 or Bek et al., 2018) since it provides a reasonable approximation of the target distribution. There are also results showing convergence in Hellinger distance of the Laplace approximation to the posterior (Schillings et al., 2020).
> ***
> * *It is well-known that as dimension of the netwrok increases or dataset grows larger the computational cost of Laplace approxiamtion and imporatnece samling can be significant. How is the relationship between effective sample size, variabaility of the weights and growth of the netwrok, dataset, and dimentionality justified?*
>
> We are unfortunately not entirely sure what is meant with this comment, but we will be happy to discuss during the discussion period.
> ***
> Given the response comments above, and that many of the concerns were already addressed in the original submission, we would appreciate a reconsideration of the perceived weaknesses of the paper and hopefully an adjustment of the score accordingly.

---

> > ### Comment · Reviewer_sMvJ · 2023-08-14
> >
> > I extend my appreciation to the authors for their response. Despite recognizing the significance of the problem, my stance remains unchanged—I believe that this work lacks the requisite novelty to warrant publication at NeurIPS, as some of my initial concerns persist. Consequently, I maintain my original score.

---

### Official Review · Reviewer_uGD4 · 2023-07-04

**Soundness:** 4 excellent
**Presentation:** 3 good
**Contribution:** 3 good
**Rating:** 7
**Confidence:** 4

**Summary:**

The paper proposes methodology to perform Bayesian inference on the (hyper) parameters of a so-called Gaussian Process Networks (GPNs), which are sets of functional equations with Gaussian Process (GP) priors on the functions relating a variable to its parents, as well as inference on the graph structure and graph posterior expectations.
Previous works in the GPN literature that consider a posterior over graphs did not put priors over hyperparameters of the GP kernels and perform associated posterior inference.
To address the challenges, the authors crucially exploit the decomposability property of p(G | D): " scores " (conditionals of X_i given PA_i ) only need to be recomputed if the parents have changed, so that scores can be reused across different graphs.
Since scores are intractable integrals over (hyper) parameters of the GPs, the authors use bridge sampling to approximate the integrals, and make use of Laplace approximations  in a two-step procedure to reduce computational cost compared to running MCMC directly with the estimated scores.
The methodology is evaluated on a good set of synthetic and real-data examples.


**Strengths:**

Overall a technically solid paper on an interesting topic for the NeurIPS community. Main strengths are as follows:
- GPNs are gaining more attention in particular due to their application in causal structure learning (von Kügelgen et al 2019), so the authors address a relevant problem of performing "fully" Bayesian inference with these models
- The scope and objective of the paper and proposed methods are clear and targeted
- The paper demonstrates an excellent knowledge of the surrounding relevant literature on MCMC for structure learning, and describes the relationship with the current work clearly
- Well explained and solid experimental section



**Weaknesses:**

- [**Presentation of Section 3**] The main weakness of the paper is the presentation of the approximations in Section 3. Even as a reader well familiar with bridge sampling, Laplace approximations and MCMC, the part from lines 143 to 172 is unclear at first read and too condensed. I am guessing the final goal is to approximate p( \Phi | D) of Eq (13). Rather than start from the various approximations needed to get there, I would suggest starting from there and write down the integrals that arise, then explain how to approximate each step. Otherwise, while I'm reading I keep thinking "Why am I approximating the scores ? What is the final goal ?". Some related questions/comments about clarity of this part below.
    - Is the reason that you approximate the scores both **(a)** via Laplace  **and**  **(b)** via bridge sampling because  **(a)** is used to build the proposal while  **(b)** is needed to compute the (self-normalized) estimator in Eq. (13), specifically, the unnormalized target ? If so you should note that the estimator in Eq (13) has an **additional bias** ( in addition to the standard bias coming from self-normalized IS)  due to the fact that the bridge sampling estimator is not unbiased. You should also state that because you are using self-normalized IS, you do not need to know the normalizing constants of either q(G|D) or p(G|D), as the ratio of these two constants cancels out.
- [**Difficulty introduced by the additional marginalization**]: while usually people assume there is inherent interest in a "fully" Bayesian treatment and marginalize hyperparameters as well when feasible (in this case, kernel hyperparameters), the authors should make it clearer why it is beneficial to do so specifically in the setting of GPNs with interest in p(G|D). There is a comment about this in lines 290-293 but it is quite vague.

**Questions:**

- How is the Gaussian proposal for the numerator of Eq. (11) chosen ? I would have expected a Laplace - IS here too ? Perhaps clarify what is the choice made here and possibly why.
 - Could you comment theoretically/intuition-wise/experimentally on which of these approximations are the ones that affects accuracy most and how ? For example, should we focus on getting better proposals $q(G|D)$ or in approximating the scores with MC (bridge sampling, in your case) more efficiently to approximate the target (numerator of the IS weights, in IS terminology) better ? Further, could you provide a couple of example $\Phi$ of interest already in the methodology (Section 3) for concreteness ?
    - The comment in lines 290-293 is important in my view, but should be expanded upon. Why/how does marginalizing the hyperparameters mitigate the bias of the Laplace approximation? Why can we not "just" not marginalize them and simply use more MCMC samples ?
- Regarding score equivalency, could you clarify why it sounds like it's not good if DAGs from the same MEC obtain different scores ? Does the property make it harder to approximate the true $p(G|D)$ somehow ?
- Does any Bernstein von Mises - type result hold for $p(G|D)$ ?


**Limitations:**

The authors do not have an explicit limitations subsection, although limitations are not hard to infer from the text, including mainly computational cost of marginalizing the hyperparameters, and the unclear effects on the performance of the approximations performed by bridge sampling; it is also not clear whether the authors can use kernels appropriate for with non-stationary functions.

---

> ### Author Rebuttal · Authors · 2023-08-09
>
> We would like to thank the reviewer for their constructive comments and interesting questions.
> * *How is the Gaussian proposal for the numerator of Eq. (11) chosen ? I would have expected a Laplace - IS here too ? Perhaps clarify what is the choice made here and possibly why.*
>
> The proposal function g() was set to a normal distribution, with its first two moments chosen to match those of the posterior distribution. This is a standard choice in the literature and the default in the bridge sampling implementation that we used (Gronau et al., 2017). We will add a clarification of this when discussing implementation details in Section 3.1.
> ***
> * *Could you comment theoretically/intuition-wise/experimentally on which of these approximations are the ones that affects accuracy most and how ? For example, should we focus on getting better proposals $q(G|D)$ or in approximating the scores with MC (bridge sampling, in your case) more efficiently to approximate the target (numerator of the IS weights, in IS terminology) better ?*
>
> This is an interesting and relevant question which is however challenging to address in all generality. Improving the proposal distribution can provide more uniform weights, leading to a more efficient exploration of the posterior and reducing variance. More accurate proposals would however require more computational power and could undermine the point of having a proposal distribution in the first place. Improving the MC estimates of the scores would on the other hand reduce the bias deriving from the bridge sampling approach (as you noted in your comments). The approximations and trade-offs used in the experiments seemed to perform quite well, though determining the best approach to perform posterior inference would be an interesting area of experimentation in this domain.
> ***
> * *Further, could you provide a couple of example $\Phi$ of interest already in the methodology (Section 3) for concreteness ?*
>
> Three examples of $\Phi$ are already provided in Section 2.1, lines 61-62.
> ***
> * *Why/how does marginalizing the hyperparameters mitigate the bias of the Laplace approximation?*
>
> The Laplace approximation is meant as an approximation of the score function, since it does not fully take into account the uncertainty regarding the hyperparameters, and only provides a good approximation if the posterior over the hyperparameters is approximately Gaussian.
> Marginalizing the hyperparameters on the other hand provides an accurate estimation of the score because it directly follows the definition of the score in equation (10). This process of marginalizing is however computationally costly, leading us to use the Laplace approximation in the first step of our approach. Finally we correct the approximation with importance sampling.
> ***
> * *Does any Bernstein von Mises - type result hold for $p(G|D)$ ?*
>
> While we are not aware of any such result, we agree it would be interesting if any such results could be established and an interesting line of research.
> ***
> * *Regarding score equivalency, could you clarify why it sounds like it's not good if DAGs from the same MEC obtain different scores ?*
>
> Two DAGs from the same MEC are theoretically indistinguishable from observational data. The lack of score equivalence in our case is caused by the GPN assumption, which introduces asymmetry into the factorizations of the joint distribution via the structural equation model (8). Because score equivalence is a common, and in some cases desirable, property to avoid strong distributional assumptions, we simply wished to emphasize that the identifiability of otherwise score equivalent DAGs is a consequence of the assumptions underpinning the GPN model.
> ***
> * *[...] it is also not clear whether the authors can use kernels appropriate for with non-stationary functions.*
>
> While we didn’t explicitly mention non-stationary kernels, the method is applicable to any kernel function as long as a marginal likelihood can be computed.

---

### Official Review · Reviewer_tnR2 · 2023-07-13

**Soundness:** 2 fair
**Presentation:** 2 fair
**Contribution:** 2 fair
**Rating:** 6
**Confidence:** 4

**Summary:**

This paper proposes an MCMC algorithm for Bayesian inference for Gaussian Process Networks where you model the distribution of a node as a function of its parents plus noise, with the function a Gaussian process.

The sampler uses a Laplace approximation to make informed moves between Networks.

The sampler is demonstrated on a simulated and real example.

**Strengths:**

This is a fine paper -- which presents a sensible method for a general learning problem. It compares with some other methods and empirically gives some evidence of better performance.

The presentation is generally good.

**Weaknesses:**

The ideas come across as rather incremental -- taking standard ideas and putting them together.

The question then becomes how important and useful is the resulting method. Here is where I think the paper is a bit lacking. The empirical study is somewhat limited. Also, with their method there a numerous tuning parameters (i.e. the specification of the prior) -- and it is unclear how robust their results are to different choices of the priors. It would also be good to see some investigation of when and why the method works well relative to other methods. I can see that the Bayesian paradigm potentially gives advantages in terms of averaging over model uncertainty and being able to quantify uncertainty -- but these come with certain disadvantages (potential dependence on prior; extra computational cost; how reliable are measures of uncertainty when the model is incorrect).

The results could be a bit clearer -- e.g. more information captions; make it clearer the E-SHD is smaller for better methods; perhaps clearer about which method is which.

Is there any code available for the method?

**Questions:**

It would be helpful to see the robustness of the results of your method as you vary the prior.

It would be good to see more thorough evaluation of the methods -- in particularly looking at understanding when the method works well/when it does not.

It would be helpful to have (easy to use) code available and details of how to reproduce the results.

---

> ### Author Rebuttal · Authors · 2023-08-09
>
> We would like to thank the reviewer for their comments, and acknowledge the time they took to assess the paper.
> * *It would be helpful to see the robustness of the results of your method as you vary the prior.*
>
> The priors we use for the hyperparameter set are standard, non-informative priors (see for instance Gramacy & Lee, 2009). Nevertheless, we agree that it would be interesting to see to what extent the results are dependent on the choice of prior. In Figure 2 of the new additional pdf in the global response we compare the results in terms of E-SHD for our GP score-based approach for different prior distributions on the length-scale hyperparameter $\theta$. In addition to the default IG(2,2) prior, we explore a IG(1,3) prior that favors higher lengthscales, as well as a IG(3,1) prior favoring lower values of $\theta$. The results show that varying the prior hyperparameter has a limited effect on the results.
> As for the prior over the graphs, we explore different choices when generating the ROC-like plots in Figure A.2 in the supplementary material.
> ***
> * *It would be good to see more thorough evaluation of the methods -- in particularly looking at understanding when the method works well/when it does not.*
>
> In our simulations, we look at different levels of non-linearity for specifically this purpose - comparing the performance of our approach under situations when linear models such as BGe are expected to work better.
> ***
> * *[...] make it clearer the E-SHD is smaller for better methods*
>
> We agree this could provide additional clarity, and will add a short comment under equation (15) mentioning this, though, as we previously noted in the manuscript, the E-SHD is not a direct measure of distance; the true posterior has a non-zero distance, so overly small values are not best.
> ***
> * *Is there any code available for the method?*
>
> As mentioned in lines 187-188, code to implement the method and reproduce the results of Sections 4 and 5 is publicly available.
> ***
> We hope the above comments have addressed the reviewer’s concerns and will be happy to hear back from them.

---

> > ### Comment · Reviewer_tnR2 · 2023-08-16
> >
> > Thanks for the reply and additional results. I am happy to see some evidence of robustness, and have increased my score on the paper to a marginal accept.

---

> > > ### Author Response · Authors · 2023-08-18
> > > **Thank you**
> > >
> > > Thanks for your positive feedback, we’re glad our response and the additional results about robustness were appreciated.

---

### Official Review · Reviewer_JYiJ · 2023-07-14

**Soundness:** 4 excellent
**Presentation:** 4 excellent
**Contribution:** 3 good
**Rating:** 7
**Confidence:** 4

**Summary:**

Within the broader task of causal network structure identification from observational data, i.e. finding the probabilistic graphical model which best explains (has highest likelihood) observations of a set of variables, this paper focus on a type of network known as GPN. GPNs are adapted to scenarios where all variables are continuous, and are built from a simple building block in which every child random variable's distribution, conditional on its parents, is a GP regression with independent Gaussian noise. For this task and this type of network, the paper proposes a fully Bayesian inference procedure based on MC and MCMC to sample PGM structure (i.e. edge presence and direction), and compute graph-related statistics of interest (such as the probability of presence of an edge between two given variables). This procedure holds the promise of more accurately representing the graph structure distribution than score-based methods.

**Strengths:**

Originality: The paper's approach consists of "pushing through" with a strong MCMC sampling procedure for the stated problem. The problem statement is not original per se, but there is moderate inventivity in putting together the required technical steps to overcome each hurdle: speeding up eq 5 (graph posterior) thanks to eq 12 (Laplace approximation of per-variable score), importance sampling for feature posteriors in eq 13. Originality is not the paper's strong suit however.
Quality: Derivations and implementations are all correct. Experimental validation on synthetic data is a necessity, especially where full enumeration of network structures is needed; the validation on the Sachs dataset is interesting, though one would wish for further real-data experiments. I appreciate that score equivalence/graph identifiability is addressed thoroughly. Evaluation methods and prior choices seem well-motivated and comparable to other literature.
Clarity: The paper is exceptionally clear. Citations are almost all accurately chosen (with the exception of a few in the introduction, cf below).
Significance: The paper tackles its stated problem heads-on. I believe we will see further Bayesian takes on network structure inference, maybe with smarter graph posterior approximations or even variational approaches for certain types of networks. The choice of GPN (and the particular further choices made for experimental evaluation) is rather flexible as it supports varied functional dependencies of child upon parent variables. The experimental evaluation places variants (order/partition) of the paper's methods at better or on par with BGe, and better than other methods. It is clearly superior in its sampling behaviour (fig 2). Its running time, unfortunately hidden in the supplementary material fig A8, is 30x that of BGe however, which might restrict applicability.

**Weaknesses:**

Significance, as discusse above.
More real-world experiments would help assess the impact, in particular if they were analysed with the level of detail of synthetic experiments sec4.

Minor errors and typos I found
- line 162: perform, not make
- line 208: data is generated
- line 266: define CPDAG
- line 289: increase, our approach
- lines 24 and 29, citations seem quite cherry-picked and do not provide a variety of viewpoints as I would expect; I would hope there are no biased self-citations here.
- supplementary line 479: do you mean TP instead of TN? eq 19, what does the lowercase p stand for in FPRp?
- table 1: I would add the usual arrow-up, arrow-down to indicate "higher/lower is better" for easier reading

**Questions:**

- To make the point lines 168 ff., would it be worth counting the number of score evaluations?
- Do some real-world scenarios require a different choice from the uniform prior over graph structures (line 141) ?
- Table 1, rightmost column: can the poor performance of your method be explained by the fact that there is no reason to predict edge absence, when absence of correlation can be captured with a fairly wide conditional distribution, i.e. low covariance in GP?

**Limitations:**

The limitations tied to algorithm choices are well discussed. It is worth motivating better why it is reasonable to focus on GPNs with respect to real-world needs, where all variable might not be continuous, where GP conditional distributions might be too loose, or where neural-network-parameterized conditional distributions might provide a more adequate model.

I see no interesting limitations to discuss related to applications or societal impact.

---

> ### Author Rebuttal · Authors · 2023-08-09
>
> We thank the reviewer for their constructive comments and suggestions, and will incorporate them into the manuscript.
> * *Minor errors and typos I found [...] lines 24 and 29, citations seem quite cherry-picked and do not provide a variety of viewpoints as I would expect; I would hope there are no biased self-citations here.
> supplementary line 479: do you mean TP instead of TN? eq 19, what does the lowercase p stand for in FPRp?*
>
> Thanks for spotting the typos! The citations are based on papers performing Bayesian analyses with Bayesian networks, and while Bayesian networks are wide-spread, Bayesian inference for Bayesian networks is less so. For example, citation 25 is the only Bayesian network paper cited in a recent review of Bayesian statistics (doi.org/10.1038/s43586-020-00001-2). The lowercase p means we normalise by P rather than N so that FPs and FNs count the same in the comparisons.
>
> * *To make the point lines 168 ff., would it be worth counting the number of score evaluations?*
>
> This would indeed be an interesting additional result. Figure 1 in the additional pdf in the global response shows the number of scores that were evaluated for the same simulations of Figure 1 in the main text. In red is the number of scores computed via the Laplace approximated posterior (“Laplace approximate”), in light green the number of additional scores evaluated via bridgesampling for the partition MCMC algorithm (“Bridgesampling, partition”) and in dark green the same quantity for order MCMC.
> Naively running the MCMC algorithm without our importance sampling approach would require computing the number of scores in red with the computationally expensive bridge sampling method. On the other hand, our importance sampling approach requires computing via bridge sampling only the number of scores shown in light/dark green, leading to much more efficient procedure.
> ***
> * *Do some real-world scenarios require a different choice from the uniform prior over graph structures (line 141) ?*
>
> Yes, and knowledge of the underlying graph can be easily incorporated into the graph prior. For example, in the real world data example in Section 5, prior knowledge of the edges that were experimentally validated by Sachs et al. (2015) would normally be included by up- or down-weighting the edges in question. Of course, to avoid circular reasoning we didn’t use them in the experiments on the Sachs data, and in fact in all of our experiments (unless stated otherwise) we employ a uniform prior when examining the performance of our approach. For real-world analyses typically one chooses a non-uniform prior informed by the setting, and indeed the cited application references [1,25,30] all have this (using either a sparsity cut-off, information from a separate biological database, or excluding edges on the basis of temporal restrictions, respectively).
> ***
> * *Table 1, rightmost column: can the poor performance of your method be explained by the fact that there is no reason to predict edge absence, when absence of correlation can be captured with a fairly wide conditional distribution, i.e. low covariance in GP?*
>
> Edge absence implies a conditional (including marginal) independence relation between the two variables in question. However, edge absence implies neither zero correlation nor zero partial correlation in general (see for example A. Vargha, 1996). Predicting edge absence can therefore be more challenging, and we agree that the absence of an edge in the “ground truth” network could have different interpretations. In particular we find stronger evidence for an edge when we allow non-linear relationships than in the linear cases, which could be an indication of more complex dependencies. Of course we didn’t want to make too strong statements, but this aspect may be worth discussing in more detail and we could adapt accordingly.

---

> > ### Comment · Reviewer_JYiJ · 2023-08-18
> > **Remaining concerns**
> >
> > * On the issue of citations: to clarify, my argument is that the choice of citations in the passages I discussed stems from always the same set of authors (mainly Moffa and Kuipers), and that I see no reason for this particular selection. I have consulted the review cited by the authors in their rebuttal (doi.org/10.1038/s43586-020-00001-2) and see that indeed, [25] is the only citation this paper features on Bayesian networks -- this does not convince me that it is a good, representative choice.
> > Bayesian network structure learning is a fairly wide topic, so it is easy to find a contrast, for example a review of Bayesian network structure learning (Kitson 2023) contains a wealth of citations from which to pick, many much more relevant that the ones proposed.
> > To further clarify, I disagree with the three citations suggestions made by fellow reviewer sMvJ, and agree with the authors in pointing out that they are not relevant to the paper's topic.
> > I remain concerned about the point I raised.
> >
> >     Kitson, N.K., Constantinou, A.C., Guo, Z. et al. A survey of Bayesian Network structure learning. Artif Intell Rev 56, 8721–8814 (2023). https://doi.org/10.1007/s10462-022-10351-w
> >
> > * I agree with the clarity issues raised by reviewer uGD4, which has been addressed neither by the authors' response, neither by the revised paper version https://openreview.net/pdf?id=bBIHqoZ3OR.
> > * The paper should more clearly state the proposed method's computational cost; as I and others pointed out, it seems to be relegated to Additional material and a brief mention line 320.
> > * Overall, I feel that several of the questions raised by reviewers, beyond a discussion here, would improve the paper if incorporated; e.g. the clarification to reviewer uGD4 on on why score equivalence is/is not desirable.
> >
> > My other questions have been addressed. Before reconsidering my score, I will await further responses by the authors, a.o. on the point regarding choice of citations.

---

> > > ### Author Response · Authors · 2023-08-20
> > > **Clarification about intended amendments to the manuscript**
> > >
> > > Thanks for your comment and for opening a discussion with us.
> > >
> > > For the main comment, despite their name, there is nothing inherently Bayesian about Bayesian networks, and we were trying to focus on fully Bayesian analyses with Bayesian networks.
> > >
> > > In response to sMvJ, we expressed our intention to make a clear distinction between Bayesian and non-Bayesian analyses and to better contextualise the work in the literature, a point we should have also explicitly mentioned in our response to you. To this aim, we will modify the current text to emphasise its allusion to fully Bayesian analyses. Furthermore, we will add some text and references, highlighting the different flavours of alternative approaches, explicitly touching on non-Bayesian and MAP point estimates, drawing from the examples nicely covered in the comprehensive review of Kitson et al. 2023 that you recommended (thanks for the pointer).
> > >
> > > In terms of revising the manuscript, our understanding is that the author guidelines explicitly forbid us to make changes at this stage (Quoting from the FAQ: “Can we upload a revision of our paper during the rebuttal/discussion period? No revisions are allowed until the camera-ready stage.”). Note that no text was allowed in the additional pdf, only figures/tables and captions.
> > > With expanded referencing to the wider literature of Bayesian networks and a clearer distinction between Bayesian and non-Bayesian approaches to their analyses, the current citations will make more sense. In line with the manuscript, we had taken a particularly strict standpoint where we intended both inferences over parameters and structures as Bayesian. With this understanding, among the applications quoted in the review of Kitson et al. 2023, the work of Moffa et al. 2017 is the only paper in the intersection between Bayesian analysis and Bayesian networks, with the one we cite a more recent development of the same work covering more general applications, using dynamic versions of Bayesian networks, still in a fully Bayesian sense.
> > >
> > > The other 5 applications referenced in the introduction of Kitson et al. 2023 all target a single-point estimate structure, and even if some use a Bayesian score to penalise the likelihood (since some penalisation is always required to avoid fully connected DAGs), they are not fully Bayesian since they do not account for the (posterior) uncertainty in structure.
> > > Likewise, the work of Kuipers et al. 2018 appears as the only paper in the intersection between Bayesian analysis and Bayesian networks in the Bayesian statistics review article we previously mentioned.
> > >
> > > Falling in the intersection of the two fields in two large recent surveys seems a reasonable criterion, which leads to two of the three referenced papers.
> > >
> > > With the restriction of being currently unable to share a revised manuscript in mind, we take on board further suggestions and will modify the text to focus on clarity, include the timing/computational cost more prominently in the main manuscript (there is also more space when revisions are allowed) and enhance the discussion on score equivalence. We thank the reviewer for reminding us that these are important points to further improve the manuscript and we hope we could assuage their concerns.

---

### Author Rebuttal · Authors · 2023-08-09

Following the reviewers' suggestions, we attach a pdf containing additional simulation results.

---

### Decision · Program_Chairs · 2023-09-21

**Decision:**

Accept (poster)

**Comment:**

Four knowledgeable referees support to accept. A fifth reviewer recommends rejection. Having looked at this reviewer's comments and the rebuttal, I side with the authors on two important points: 1) the paper, indeed, is related to a different problem than the one alluded to in the references provided by the reviewer; 2) the reviewer does not provide a concrete list of methods that could be used as baselines in the experiments, rather refers to a lack of comparisons against state-of-the-art approaches. Several other concerns by the reviewer were already addressed in the original submission. I therefore recommend accepting. Authors must follow the reviewer's suggestions, in particular, addressing their comments on the presentation of the approximations in Section 3.